# Cefiderocol and Sulbactam-Durlobactam against Carbapenem-Resistant *Acinetobacter baumannii*

**DOI:** 10.3390/antibiotics12121729

**Published:** 2023-12-14

**Authors:** Arta Karruli, Antonella Migliaccio, Spyros Pournaras, Emanuele Durante-Mangoni, Raffaele Zarrilli

**Affiliations:** 1Department of Precision Medicine, University of Campania “L. Vanvitelli”, 80138 Naples, Italy; arta.karruli@unicampania.it; 2Department of Public Health, University of Naples Federico II, 80131 Naples, Italy; antonella.migliaccio10@gmail.com; 3Clinical Microbiology Laboratory, Medical School, “Attikon” University General Hospital, National and Kapodistrian University of Athens, 1 Rimini Street, 12462 Athens, Greece

**Keywords:** carbapenem-resistant *Acinetobacter baumannii*, cefiderocol, sulbactam-durlobactam, antimicrobial susceptibility, mechanisms of resistance, antimicrobial therapy

## Abstract

Infections caused by carbapenem-resistant *Acinetobacter baumannii* (CRAB) remain a clinical challenge due to limited treatment options. Recently, cefiderocol, a novel siderophore cephalosporin, and sulbactam-durlobactam, a bactericidal β-lactam–β-lactamase inhibitor combination, have been approved by the Food and Drug Administration for the treatment of *A. baumannii* infections. In this review, we discuss the mechanisms of action of and resistance to cefiderocol and sulbactam-durlobactam, the antimicrobial susceptibility of *A. baumannii* isolates to these drugs, as well as the clinical effectiveness of cefiderocol and sulbactam/durlobactam-based regimens against CRAB. Overall, cefiderocol and sulbactam-durlobactam show an excellent antimicrobial activity against CRAB. The review of clinical studies evaluating the efficacy of cefiderocol therapy against CRAB indicates it is non-inferior to colistin/other treatments for CRAB infections, with a better safety profile. Combination treatment is not associated with improved outcomes compared to monotherapy. Higher mortality rates are often associated with prior patient comorbidities and the severity of the underlying infection. Regarding sulbactam-durlobactam, current data from the pivotal clinical trial and case reports suggest this antibiotic combination could be a valuable option in critically ill patients affected by CRAB infections, in particular where no other antibiotic appears to be effective.

## 1. Introduction

Bacteria belonging to the genus *Acinetobacter* are non-fermentative Gram-negative coccobacilli that emerged as an increasingly frequent cause of healthcare-associated infections and hospital outbreaks [1,2,3]. *Acinetobacter baumannii* is the most clinically relevant species and is responsible for various healthcare-associated infections, including pneumonia, bloodstream infections, urinary tract infections, and wound infections. Also, *A. baumannii* causes community-acquired infections, although to a lesser extent [2,4].

*A. baumannii* strains responsible for epidemic spread are resistant to carbapenems and show intermediate resistance to tigecycline, but usually retain susceptibility to colistin and are classified as multi-drug-resistant (MDR) or extensively drug-resistant (XDR) [1,2,4]. Therefore, *A. baumannii* infections show limited treatment options and high mortality, especially in critically ill patients [2,4], and this has led the World Health Organization to classify carbapenem-resistant *A. baumannii* (CRAB) as a “critical” pathogen among the antibiotic-resistant bacteria of global priority for the development of new antibiotics [5].

Recently, cefiderocol (FDC; S-649266), a novel siderophore cephalosporin, which possesses a broad activity against CRAB in vitro and in vivo [6,7,8], has been approved by the Food and Drug Administration for the treatment of serious infections caused by carbapenem-resistant Gram-negative bacteria (CR-GNB) [9]. Also, sulbactam-durlobactam (SUL/DUR), a bactericidal β-lactam–β-lactamase inhibitor combination [10], has been demonstrated to be active against CRAB in vitro [11,12,13] and in vivo [14,15]. SUL/DUR, XACDURO, was approved in the USA for the treatment of hospital-acquired bacterial pneumonia and ventilator-associated bacterial pneumonia (HABP/VABP) caused by susceptible isolates of *A. baumannii* [16].

The aim of this review is to discuss: (i) chemical structures and pharmacological data of FDC and SUL/DUR; (ii) in vitro and in vivo activity of FDC and SUL/DUR against CRAB and mechanisms of resistance; (iii) utilization and efficacy of FDC and SUL/DUR in therapeutic regimens against CRAB.

## 2. Cefiderocol Is a Novel Siderophore Cephalosporin

### 2.1. Chemical Structure and Pharmacological Data

FDC is a novel catechol-substituted siderophore cephalosporin that is structurally different from other recently developed siderophore-conjugated molecules, showing a high stability against various serine-type and metallo-type carbapenemases and ESBLs [6]. FDC consists of a 4-membered β-lactam ring bound to a 6-membered dihydrothiazine ring, which is covalently bound in the 3-position to a catechol 2-chloro3,4-dihydroxybenzoic acid moiety. The quaternized N-methyl-pyrrolidine is identical to the pyrrolidinium group found in cefepime and confers zwitterionic properties that enable it to rapidly penetrate the outer cell membrane of Gram-negative bacteria (Figure 1A) [6]. Furthermore, FDC presents in the 7-position an aminothiadiazole ring bound to oxime and dimethyl groups, which improves the stability to β-lactamases and the overall antimicrobial activity (Figure 1A) [6]. The catechol moiety is important for the siderophore function and “Trojan horse” strategy of FDC; indeed, the molecule is able to chelate ferric iron and cross the outer membrane of Gram-negative organisms using the ferric iron transport system (Figure 1). The transport of FDC from the outer membrane to the periplasmic space is mediated by passive diffusion via porin channels or active transport linked to TonB-dependent siderophore receptors PiuA and PirA [6] (Figure 1A). Thus, a positive correlation has been demonstrated between FDC susceptibility and the expression of PiuA and PirA TonB-dependent receptors (TBDRs) in *A. baumannii* [17,18]. Also, the high affinity of PiuA and PirA transporters for siderophores allows the transport of FDC in the periplasmic space even at low concentrations (Figure 1B) [19,20]. The high affinity binding of FDC to penicillin-binding proteins (PBPs), primarily PBP3, in the periplasmic space (Figure 1B) results in the inhibition of peptidoglycan synthesis and cell death [6,17,21,22].

### 2.2. In Vitro and In Vivo Activity against CRAB

FDC susceptibility testing may be performed with disk diffusion or broth microdilution (BMD).

Disk diffusion is performed with an FDC 30 μg disk according to the EUCAST standard recommendations for non-fastidious organisms [23].

As the FDC requires low levels of iron to have optimal activity, BMD MICs need to be determined in iron-depleted Mueller–Hinton broth (MHB). As the iron concentration in the MHB affects reproducibility, broth preparation requires particular care (https://www.eucast.org/eucastguidancedocuments/, accessed on 10 November 2023). Indeed, chelators used to remove iron will also remove zinc, calcium, and magnesium, which have to be added back [24].

The EUCAST issued specific recommendations for reading the BMD MICs of FDC (EUCAST guidance document on FDC BMD, December 2020). According to this document, the MIC corresponds to the first well where the reduction of growth is represented by a button of <1 mm or by the presence of light haze/faint turbidity.

The inhibition zone diameters and MICs of FDC may be interpreted according to the EUCAST non-species-specific pharmacokinetic/pharmacodynamic (PK/PD) breakpoints (zone diameter ≥17 mm, corresponding to MIC below the PK-PD susceptible breakpoint ≤ 2 mg/L, MIC > 2 mg/L as resistant; EUCAST Clinical Breakpoint Tables v. 13.1, accessed 10 November 2023) or to CLSI breakpoints (zone diameter ≥15 mm or MIC ≤ 4 mg/L as susceptible, MIC ≥ 16 mg/L as resistant) [25].

The EUCAST has evaluated several commercial tests, all of which had problems of accuracy, reproducibility, bias, and/or skipped wells, and noted that it is difficult to interpret the FDC susceptibility of isolates in the area of technical uncertainty (ATU). For this reason, a warning has been issued (www.eucast.org/ast-of-bacteria/warnings, accessed on 10 November 2023) and it is recommended that laboratories start testing FDC with disk diffusion, which, when correctly performed and calibrated, is predictive of susceptibility and resistance outside the ATU. Within the ATU, the EUCAST recommended ignorance of the ATU and interpretation using the zone diameter breakpoints in the breakpoint table. Despite the investigation of several products by the EUCAST, the existing issues have not been resolved, and it was decided that the warning should still remain.

Some recent evaluations of FDC susceptibility tests for *Acinetobacter baumannii* isolates will be presented herein. Liu et al. [26] observed that disk diffusion was difficult to assess, but the limited isolates that actually exhibited resistance by BMD (CLSI breakpoint) were categorized as susceptible by disk diffusion. Jeannot et al. [27] evaluated two commercial BMD methods and discs from three manufacturers, compared with the reference BMD. They showed that none of the tested methods met the accuracy requirements. Both BMD methods exhibited acceptable categorical agreement rates. MIC gradient testing was strongly discouraged, and disc diffusion could be used to screen for susceptibility, setting a critical diameter of 22 mm. Finally, a quick test, the rapid FDC *Acinetobacter baumannii* NP test for the detection of FDC susceptibility/resistance in *A. baumannii*, was recently proposed [28], and evaluated in comparison with the BMD reference method. The test showed very high sensitivity and specificity, obtained within 4:30–4:45 h of incubation, and had only a single very major error, using an isolate with the MIC of 8 mg/L. Shortridge et al. [7] reported that the susceptibility of meropenem-resistant *Acinetobacter* spp. (306 isolates) to FDC was 97.7% using CLSI criteria. Despite the susceptibility testing used, FDC showed excellent in vitro activity against *Acinetobacter* spp., ABC, and CARB [6,7,26,27,28].

In keeping with the in vitro activity of FDC, Matsumoto et al. [8] demonstrated the efficacy of FDC against CRAB in an immunocompetent rat respiratory tract infection model, recreating human plasma pharmacokinetics. Also, Gill et al. showed effective in vivo bactericidal activity of cefiderocol in combination with ceftazidime/avibactam or ampicillin/sulbactam using human simulated regimens against all the cefiderocol non-susceptible *A. baumannii* isolates tested [29].

### 2.3. Mechanisms of Resistance

FDC resistance has been associated with the reduced expression of the siderophore receptor *pirA* and *piuA* genes in several *A. baumannii* strains [17,18]. Also, Malik et al. showed that two mutations in the midst of a beta strand of PirA diminished the functionality of this receptor protein in two FDC-resistant *A. baumannii* strains [17]. Moreover, efflux-pump systems may underlie resistance to FDC. Liu et al. showed that mutations of the BaeS (D89V) and BaeR (S104N) two-component system regulator increased FDC MIC, and this effect was mediated by the up-regulation of MFS and the MacAB-TolC efflux pumps (Figure 1B) [30].

Mounting evidence indicates that resistance to FDC in *A. baumannii* is mediated by the production of PER- and NDM- β-lactamases [31,32]. Specifically, Poirel et al. demonstrated that: (i) *A. baumannii* isolates positive for PER-1-7 or NDM-1-5-9 β-lactamases showed increased MIC for FDC; (ii) the transformation of *E. coli* and *A. baumannii* CIP70.10 with recombinant plasmids producing PER-1 or NDM-1,5,9 β-lactamases increased FDC MIC; (iii) β-lactamase crude enzymatic extracts from PER-1-producing recombinant *E. coli* strains and to a lesser extent from NDM-1-producing recombinant *E. coli* strains showed a significant hydrolysis rate of FDC [31]. Also, Liu et al. demonstrated that inactivation of the *bla*_PER-1_ gene through allelic replacement restored the susceptibility to FDC in *A. baumannii* XH740, and that the phenomenon was reverted by the introduction of PER-1 into the knockout strain [32]. In addition, Asrat et al. demonstrated that the over-expression of *bla_ADC_* subtypes β-lactamases correlated with elevated FDC resistance, and that site-specific insertional inactivation of blaADC–25 or blaADC-33 increased FDC susceptibility in *A. baumannii* strains (Figure 1B) [18]. Finally, Ile236Asn and His370Tyr mutations were found in PBP3 from one FDC-resistant *A. baumannii* isolate [17].

FDC has also been reported to exhibit heteroresistance [33,34,35]. The population analysis profiling (PAP) of CRAB isolates showed the change to the non-susceptible phenotype after exposure to FDC in 1 of 10 isolates and the occurrence of heteroresistance in 8 of 10 isolates [33]. The occurrence of heteroresistance after FDC exposure and the relationship between heteroresistance and clinical outcomes were recently evaluated in CRAB isolates from the CREDIBLE-CR study [34]. By using the PAP, only 7/38 CRAB isolates were susceptible, 18/38 were heteroresistant, and 13/38 were resistant. Heteroresistance, however, was not related with worse clinical or microbiological outcomes compared to non-heteroresistant isolates [34].

In contrast with FDC, which is generally active against CRAB, none of the newer b-lactam/b-lactamase inhibitor combinations, including ceftazidime–avibactam, imipenem–relebactam, meropenem–vaborbactam, and ceftolozane–tazobactam, retain activity [7]. For this reason, FDC has not generally been reported to exhibit cross-resistance with these antimicrobial classes against CRAB.

### 2.4. Therapy against CRAB

#### 2.4.1. Studies Evaluating the Clinical Efficacy of FDC

##### Randomized, Phase III Studies

The efficacy of FDC has been evaluated in two pivotal, randomized, multi-center, phase 3 clinical studies:APEKS-NP—enrolling patients with nosocomial pneumonia;CREDIBLE-CR—focusing on severe infections caused by CR Gram-negatives.

The APEKS-NP [36] was a double-blind, randomized, active-controlled, non-inferiority study, enrolling adult patients with documented nosocomial pneumonia due to Gram-negative bacteria. Exclusion criteria were CR pathogens, Gram-positive or anaerobic pathogens, viral, atypical, chemical, or community-onset pneumonia. A total of 292 patients were included in the study after modifying the randomization group (comprising a total of 300 patients) for exclusion criteria, of whom 148 patients received treatment with FDC and 145 with meropenem, both at a dose of 2 g every 8 h intravenously. All patients received concomitant treatment with linezolid. Infections due to *A. baumannii* were present in 47 patients (16%), equally divided among the two study groups.

Although CR pathogens were excluded from the study, 56 patients (19%) were eventually found to have a CR infection after randomization, mostly due to *A. baumannii* and *Acinetobacter* spp.; these patients were still included in the study analyses.

A clinical cure was achieved in 65% of patients in the FDC group compared to 67% of those in the meropenem group.

This trial also analyzed endpoints based on individual pathogens: FDC was non-inferior to meropenem in infections due to *A. baumannii*, with clinical cure rates of 52% and 58%, respectively. The microbiological eradication of *A. baumannii* was achieved in 39% of cases treated with FDC compared to 33% of those treated with meropenem. Also, similar mortality rates were seen in the two arms of treatment for *A. baumannii*, with mortality at 14 days being 19% in the FDC group and 22% in the meropenem one [36].

These findings appear substantially different from those obtained in the other study with FDC, the CREDIBLE-CR trial [37]. This was a multicenter, randomized, open-label clinical study of FDC versus the “best available therapy” (BAT). Inclusion criteria were the identification of a carbapenem-resistant Gram-negative organism in a clinical sample or prior antimicrobial treatment failure. In complicated urinary tract infections (cUTI), FDC was used solely as monotherapy, whereas for other infection syndromes, another anti-Gram-negative antibiotic along with FDC was allowed. For the BAT arm, up to three antibiotics combined were allowed, with only 29% of the BAT group patients receiving monotherapy. Most treatment regimens were based on colistin, which was administered to 66% of patients overall. FDC was given at a dose of 2 g every 8 h and was mostly used as monotherapy (83% of patients).

A total of 152 patients were randomized in CREDIBLE, of whom 101 received FDC and 49 BAT. *A. baumannii* was the causative pathogen in 54 of 118 carbapenem-resistant microbiological intention-to-treat (ITT) subjects (46%). Indeed, *Acinetobacter* was the most prevalent CR pathogen at baseline in the FDC group (65% in FDC vs. 53% in the BAT group). Considering all infection syndromes, clinical cure rates at the test of cure (TOC) for patients who received FDC treatment compared to those who received BAT were 53% versus 50%, respectively, with clinical failure rates equal to 34% versus 37%, respectively. Clinical cure rates at TOC were higher for cUTI compared to other infection syndromes. Clinical cure and microbiological eradication rates along with the counterpart negative outcome based on different types of infections are described in detail in Table 1.

Interestingly, all-cause mortality in the subgroup of patients with *A. baumannii* infections at the end of treatment with FDC was 49%, compared to only 18% with BAT. Most of the deaths occurred either early, within the first 3 days, or late, from day 29 until the last follow-up. In the period between these two time points, deaths were similar between the two groups, suggesting that factors related to the underlying condition at randomization for the first 3 days and other complications developed after day 29 could have influenced the outcome.

It should be underscored that patients with infections due to *Acinetobacter* spp. who received treatment with FDC were overall more complex, showing a higher prevalence of baseline moderate/severe renal dysfunction, ICU admission at randomization, ongoing shock, or shock within 31 days before randomization, compared to those with the same pathogen treated with BAT [37]. Moreover, as the authors stated in their data discussion, the 28-day mortality rate in the FDC group of the CREDIBLE study was similar to that of other studies comparing the efficacy of colistin monotherapy vs. combination therapy for *A. baumannii* infections [38,39,40,41].

##### Clinical Studies Comparing the Efficacy of FDC with That of Colistin

Clinical trials provide the first key insights into the effectiveness of new drugs and represent the gold standard in evaluating the efficacy and safety of novel antimicrobials. However, clinical trials may be limited by the selected population studied; with all the exclusion criteria, they may leave out most of the patient subgroups encountered in daily clinical practice.

While formal clinical trials achieve mostly internal validity, other clinical/real-world studies are the ones that reach external/generalized validity [42]. Various studies, not fulfilling criteria for randomized controlled trials, have assessed the efficacy and safety of FDC compared to other treatments. Six single-center and one multi-center [43,44,45,46,47,48,49] retrospective and prospective observational studies evaluated the efficacy of FDC compared to a colistin-based regimen or other antibiotics, with a range of 73–124 patients included, and mostly ventilator-associated pneumonia (VAP) or bloodstream infections (BSI) due to carbapenem-resistant *Acinetobacter baumannii* (CRAB). FDC and colistin were used both as monotherapy and in combination regimens in six studies [43,44,45,46,47,49], whereas Pascale et al. [48] used FDC monotherapy.

##### Mortality

The FDC treatment was non-inferior compared to colistin in terms of mortality, which ranged from 10% to 55% (Table 2). Only in the Mazzitelli et al. study, FDC had a mortality which was higher compared to colistin (51% vs. 37%) [46]. As in the CREDIBLE-CR study, mortality or clinical failure was higher in more comorbid patients or those with sepsis/septic shock, along with higher SOFA scores [43,44,45,46,47,48,50,51,52]. Indeed, mortality associated with *A. baumannii* infections was shown to be higher in patients whose infections were complicated by septic shock [53,54,55,56].

##### Mortality Based on Type of Infection

Falcone et al., Russo et al., Dalfino et al., Rando et al., and Bavaro et al. showed lower mortality in the FDC group compared to colistin [43,44,45,47,49]. This difference was seen also in patients with BSI [43], with the FDC regimen translating into lower 14-day and 30-day mortality rates of 7.4% and 25.9% vs. 42.3% and 57.7% with the colistin-based regimen. Interestingly, there was no difference in 14- and 30-day mortality for the two groups among patients with VAP. Indeed, in another study [51], FDC was shown to be associated with a higher rate of clinical success in BSI (75%) compared to respiratory infections (45.8%). Similarly, Palermo et al. [57] observed a mortality of 61.5% in VAP compared with 46.7% in BSI due to CRAB.

##### Other Outcomes

In the Falcone et al. and Rando et al. studies, microbiological failure was higher with FDC compared to colistin (17.4% vs. 6.8% and 53% vs. 31%, respectively) [43,47]. In contrast, in the other three studies, microbiological failure was similar or higher in the colistin group [45,46,48], and in the Russo et al. study, microbiological failure was associated with higher mortality in the colistin group [44]. Clinical cure rates were also higher in the FDC group compared to colistin [45,46,48,49].

Adverse events were mostly seen in colistin-treated subjects [43,49]. Regarding treatment-emergent acute kidney injury (AKI), Pascale et al. and Dalfino et al. showed similar rates in both groups, whereas Rando et al. and Mazzitelli et al. observed this adverse event mostly in the colistin group [45,46,47,48,49], which is one of the major concerns for this antibiotic [58,59].

##### Clinical Studies and Case Series Assessing the Efficacy of FDC in Treating CRAB Infections

Two multi-center and several single-center studies (Table 3) have assessed the real-life efficacy of FDC in treating CRAB infection [50,51,52,60,61,62,63,64,65], generating similar data to the comparative studies described above. Most of the studies included multiple carbapenem-resistant Gram-negatives [52,57,60,61,62,63,65]. One study, the largest in size, focused only on *Acinetobacter* spp. infections [51], including 147 patients, of whom 146 had *A. baumannii* infections. One study focused on CRAB infections only [50]. Case series included 8 to 16 patients (with *A. baumannii*). Infection syndromes included in these studies were mostly BSI and respiratory infections. Clinical cure rates ranged from 32.5% to 77.8% [50,51,57,60,61,62,63,64,65,66]. Microbiological failure occurred in 0% to 25%. One case series on burn patients by Smoke et al. reported a microbiological failure rate as high as 88% [64]. However, in this study, the reported resistance to FDC was 60% of the tested isolates. Mortality ranged widely, from 12.5% to 51% [50,51,52,57,60,61,62,63,64,65].

Adverse events associated with FDC administration were very infrequent, being described in none to less than 5% of treated subjects. However, in the study by Piccica et al. [52], acute kidney injury occurred in 38% of cases, and in the case series by Wicky et al. [65], 6 of 9 patients experienced an adverse event, mostly encephalopathy.

Regarding the efficacy of combination treatment compared to FDC monotherapy, no significant difference was found in these studies in terms of mortality [46,50,51,52,58,62], microbiological eradication [46,50,52,62], and clinical cure [46,50,51,62].

In the case series by Gavaghan et al., mortality was 60%, with 62% of the analyzed isolates that were susceptible to FDC. In this study, the explanation for such a high mortality rate could actually be related with the observed high resistance rates to FDC [63]. Also, the study by Smoke et al. on 11 burn patients with infection due to CRAB showed not only a clinical cure rate of 36% with a high microbiological failure rate, but also a 60% rate of resistance prior to/after FDC treatment [64]. In a multi-center study by Piccica et al. on 142 CR GN infections treated with FDC, among 28 strains of *A. baumannii* with available microbiological data, 10 (35.7%) were resistant to FDC. However, in this study, resistance was not associated with a higher risk of mortality [52].

Reduced FDC susceptibility was also seen in the CREDIBLE and APEKS-NP clinical trials [36,37], with an FDC MIC increase of fourfold or more during/after treatment observed in 12 of 106 isolates in the CREDIBLE study and 7 of 159 isolates in the APEKS-NP study [67]. However, despite increasing more than fourfold, the MIC remained ≤4 µg/mL for most of the isolates. Mutations were found in only three isolates.

The in vivo emergence of resistance or reduced susceptibility to FDC was addressed also by other clinical studies. In the Falcone et al. study, 8.5% of isolates developed resistance to FDC during treatment and 50% (4/8) of patients who showed microbiological failure developed resistance to FDC during treatment [43]. In another case series by Falcone et al. [60] on 10 patients, of whom eight had an *Acinetobacter baumannii* infection, one patient with available repeated MIC values (of two who experienced microbiological failure) had a 16-fold increase in MIC from 0.25 µg/mL to 4 µg/mL [60]. The occurrence of heteroresistance in CRAB isolates has been demonstrated in CRAB isolates [33,34,35]. The presence of heteroresistance in Gram-negative isolates susceptible to FDC was found also in another study, which observed a 59% heteroresistance rate (64 of 108 *A. baumannii* isolates) with full resistance detected only in 8% [68]. Heteroresistance implies the presence of minor subpopulations of cells resistant to a specific antibiotic which remain undetectable by most antibiotic susceptibility tests and may predominate after exposure to that antibiotic, causing treatment failure or microbiological persistence. It was suggested as a possible mechanism underlying the MIC increase of isolates in the CREDIBLE study [68]. However, despite the high prevalence of heteroresistance to FDC in *A. baumannii*, its clinical impact is yet to be understood. In vivo, the emergence of resistance during treatment remains uncommon and is mostly described in low percentages/sporadic cases of the published studies [69].

Consistent with these data, the Infectious Disease Society of America (IDSA) recommended that FDC should be used with caution for CRAB infections and as part of a combination treatment regimen [70].

Other studies on a few peculiar cases in terms of site of infection described the efficacy of FDC in treating XDR/PDR infections, including one case of spondylodiscitis [71], one case of osteomyelitis, and one of prosthetic joint infection [72,73]. FDC was shown to reach optimal cerebrospinal fluid concentrations above MIC levels, achieving a clinical cure in two patients with central nervous system infections [74].

In conclusion, the growing wealth of data from several, mostly small, retrospective and single-center studies, provides somehow conflicting real-life data that do not add further robust evidence, and do not help solving the doubts raised by the CREDIBLE-CR study.

## 3. Sulbactam-Durlobactam: A Novel Combination of a Beta-Lactamase Inhibitor with Beta-Lactam Activity and a Non-Beta-Lactam/Beta-Lactamase Inhibitor

### 3.1. Chemical Structure and Pharmacological Data

Durlobactam (DUR), previously identified as ETX2514, is a non-beta lactam diazabicyclooctanone (DBO) inhibitor of class A, C, and D serine β-lactamase, but not class B metallo-β –lactamases [75]. In particular, DUR exhibits variable activity against class D beta-lactamases, with the highest activity being found for OXA-48 with respect to OXA-10, OXA-23, and OXA-24 [75]. DUR is active in its carbamylated state, i.e., when its active site serine nucleophile reacts with a β-lactamase, opening the cyclic urea in a reversible manner (Figure 2A) [10]. Sulbactam (SUL) is a semi-synthetic penicillanic acid which exhibits weak intrinsic inhibitory activity against CTX-M-15, SHV-5, TEM-1, and KPC-2 class A serine β –lactamases, but not class C and D beta-lactamases (Figure 2A) [76]. The combination of DUR with SUL restores the susceptibility to SUL in *A. baumannii* strains producing class A, C, and D β-lactamases [10,75].

The passive diffusion via porin channels (e.g., the most common OmpA porin) mediates the transport of DUR and SUL across the outer membrane. Subsequently, within the periplasmic space, SUL exerts its intrinsic antibacterial activity through the inhibition of PBP1a, PBP1b, and PBP3, but not PBP2, whilst DUR inhibits PBP2 in *A. baumannii* (Figure 2) [10]. A recent study also demonstrated that, in the presence of DUR, the FDC MICs decreased in *A. baumannii* strains producing PER-1 beta-lactamase and provided an in silico structural modeling of PER-1 binding with both FDC and DUR [32]. Based on the structural model of PER-1 binding, the authors suggested that the FDC and DUR combination might be an effective therapeutic approach against *A. baumannii* strains producing PER-1 enzymes [32].

### 3.2. In Vitro and In Vivo Activity against CRAB

Sulbactam/durlobactam (SUL/DUR) was approved in May 2023 by the USA FDA to treat hospital-acquired and ventilator-associated bacterial pneumonia (HABP/VABP) caused by susceptible isolates of the *Acinetobacter baumannii-calcoaceticus* complex (ABC) in patients 18 years of age and older [77].

For this reason, the CLSI Subcommittee on Antimicrobial Susceptibility Testing performed a breakpoint revision meeting, and new breakpoints were introduced for SUL/DUR by an ad hoc working group, and subsequently approved by the subcommittee. The proposed MIC breakpoints for ABC were susceptible (S; ≤4/4 μg/mL, ≥17 mm), intermediate (I; 8/4 μg/mL, 14–16 mm), and resistant (R; ≥16/4 μg/mL, ≤13 mm). DUR was tested at a fixed concentration of 4 μg/mL.

These breakpoints aligned with the epidemiological cut-off value, that included 98.3% of isolates tested in a large representative collection of 5,032 clinical isolates of ABC collected in 33 countries across the Asia/South Pacific region, Europe, Latin America, the Middle East, and North America between 2016 and 2021 [11]. In this study, SUL alone had a MIC_90_ of 64 μg/mL, whereas the SUL/DUR combination had a MIC_90_ of 2/4 μg/mL.

In most studies, all showing excellent activity of SUL/DUR against representative collections of ABC, MICs for SUL-DUR were determined by the CLSI standard BMD using a cation-adjusted Mueller–Hinton broth, with SUL-DUR tested in 2-fold dilutions of SUL in combination with a fixed concentration of 4 mg/mL of DUR [11,12,78]. In keeping with the in vitro data, treatment with SUL/DUR resulted in a dose-dependent reduction in XDR *A. baumannii* in both the neutropenic mice thigh abscess and pneumonia infection models [14].

### 3.3. Mechanisms of Resistance

The frequency of spontaneous in vitro resistance to SUL/DUR was low and occurred at 7.6 × 10^−10^ to <9.0 × 10^−10^ at 4 × MIC [79]. Most frequent mutations S390T, V505L, and T511A identified in stable mutants occurred in the *ftsI* gene that encoded the target of SUL PBP3 and was near the active site serine (Ser336) [79]. Mutations in tRNA synthetases, *aspS*, and *gltX* genes were also identified, and were linked to the induction of the stringent response, which rendered PBP2 dispensable [79]. Mutations A515V and less frequently T526S were identified also in the PBP3 encoding gene in the proximity to the SUL-binding site in SUL/DUR-resistant isolates from Greece, Switzerland, and France [12,13]. Additional mutations occurred in PBP1a, PBP1b, and PBP2 encoding genes in SUL/DUR resistant isolates from Switzerland and France (Figure 2) [13].

### 3.4. Therapy against CRAB

#### 3.4.1. Clinical Trials Assessing Efficacy and Safety of SUL-DUR

The efficacy of SUL-DUR in treating *A. baumannii* infections was assessed in a phase III clinical trial [80]. This was a multinational, randomized, active-controlled, non-inferiority trial. A total of 125 patients were included in the efficacy analysis, of whom 63 were treated with SUL-DUR at the dose of 2 g every 6 h, and 62 with colistin, with a maintenance dose of 2.5 mg/kg after a loading dose of 5 mg/kg of colistin base activity. Imipenem/cilastatin was used as a combination agent in both groups. Infections included BSI, hospital-acquired pneumonia (HAP), and VAP. SUL–DUR was non-inferior to colistin in terms of 28-day all-cause mortality in the microbiologically (CRAB) modified ITT population.

The 28-day all-cause mortality was 19% (12 patients) in the SUL–DUR group compared with 32.3% (20 patients) in the colistin group, with an observed treatment difference of –13.2% (95% C.I. –30 to 3.5). The 14-day all-cause mortality rate in the microbiologically (CRAB) modified ITT population was 6% (4 of 64) with SUL–DUR versus 19% (12 of 63) with colistin.

The authors performed a Kaplan–Meier analysis showing patients treated with SUL-DUR had a higher survival probability compared to those treated with colistin, with the difference becoming evident after the 6th day of treatment.

As for the microbiological eradication at the test of cure, a more favorable outcome was observed in the SUL-DUR group compared to colistin (43 of 63 treated with SUL-DUR [68%] vs. 26 of 62 patients treated with colistin [42%]).

Nephrotoxicity assessed by RIFLE criteria was significantly lower in the SUL-DUR group compared to colistin, (12 of 91 (13%) vs. 31 of 85 (38%), respectively). The overall incidence of any adverse event was 88% in the SUL-DUR and 94% in the colistin group, whereas for serious adverse events, it was 40% in the SUL-DUR and 49% in the colistin group.

#### 3.4.2. Clinical Studies/Case Reports of SUL-DUR Efficacy in Real Life

At present, there is no published real-life experience that has assessed SUL-DUR efficacy. However, a few case reports have been published (Table 4) [81,82,83].

Zaidan et al. [81] described a 55-year-old woman with pneumonia due to COVID-19, complicated by respiratory failure and on mechanical ventilation, with sputum cultures negative after extubation and clinical improvement. She later developed hypoxia again requiring mechanical ventilation. Respiratory cultures then became positive for PDR *A. baumannii* and pan-susceptible *Pseudomonas aeruginosa*. Treatment with meropenem and ampicillin-SUL (after an empirical course with meropenem and vancomycin) did not improve her clinical conditions (respiratory failure and septic shock requiring vasopressor support). Therefore, she was switched to FDC plus SUL-DUR. This treatment appeared to be successful, as her clinical condition improved within 72 h, and after a 14-day course regimen, her respiratory conditions also improved, allowing a simple tracheostomy. The patient was subsequently transferred to a long-term care facility and discharged home several weeks later.

Holger et al. [82] described a 50-year-old male patient hospitalized for pulmonary embolism with pulmonary infarction. After intubation, thoracotomy, partial decortication, and right thoracoscopy with the placement of three tubes, a bronchoalveolar aspirate was positive for *A. baumannii*. Being the strain susceptible to meropenem, the empirical treatment with piperacillin/tazobactam was switched to meropenem. The patient underwent a second thoracotomy, with right lower lobe resection (which showed necrosis, hemorrhage, and abscess) and complete decortication due to multiloculated pleural effusions. *A. baumannii* was again isolated in the pleural tissue, but with a worse antimicrobial susceptibility pattern: intermediate to colistin and with a tigecycline MIC of 2 mg/L. Treatment was initially switched to tigecycline and subsequently to colistin plus meropenem due to deteriorating clinical conditions. In light of the side effects of colistin and defined susceptibility to FDC, treatment was switched to FDC. Respiratory samples remained positive for *A. baumannii* over 45 days, after which the isolate became resistant to FDC. The patient was then switched to eravacycline. Clinical conditions worsened again, with increasing eravacycline MIC, and treatment was switched to FDC + tigecycline. Due to the persistence of purulent secretions from the chest tube, the MIC of SUL-DUR was assessed, being 8 mg/L and decreased to 4 mg/L after meropenem addition. After a 2-week treatment course with SUL-DUR at a dose of 2 g every 6 h plus meropenem, no further output from the chest tube was seen. SUL-DUR maintained activity after the last debridement and no adverse events were observed. The patient was subsequently discharged.

Tiseo et al. [83] described treatment with SUL-DUR in a young woman who suffered a severe burn injury with 45% of body involvement. She developed a central line BSI due to CRAB isolate, which showed resistance to FDC and susceptibility to colistin. She was treated with colistin and tigecycline. CRAB and *P. aeruginosa* were isolated from skin lesions. She developed a kidney injury and started continuous renal replacement therapy. Her respiratory function worsened with bilateral multiple consolidations, and a bronchoalveolar lavage fluid became positive for CRAB resistance to FDC and colistin and for carbapenem-resistant *P. aeruginosa*. The MIC for SUL/DUR was 1.5 mg/L. Therefore, she was switched to SUL-DUR combined with colistin for the carbapenem-resistant *P. aeruginosa*. After 12 days of treatment, her respiratory function improved, and CRAB was eradicated from all sites of isolation. The patient was alive at 30 days.

In conclusion, with the possible selection bias of positively evolving cases, SUL-DUR appeared as an effective salvage treatment in a very limited number of clinical case reports.

## 4. Conclusions and Future Directions

Appropriate antibiotic administration is lifesaving in critically ill patients, and its judicious use is vital to keep the antimicrobial resistance risk as low as possible in our communities. This is particularly true for novel antibiotics that are being specifically developed to tackle CR Gram-negatives, including *A. baumannii*. The data we presented clearly urge clinicians not to use FDC and SUL/DUR for non-severe infections or when less potent options are appropriate.

In the frame of strategies to control MDR infection spread, the reduced use of antibiotics appears crucial. Combination antimicrobial regimens are very commonly used in an attempt to control DTR infections and reduce mortality, and this is particularly relevant for CRAB infections [84]. However, combination therapy inherently increases antibiotic usage, possibly perpetuating antimicrobial resistance spread. In addition to several clinical studies showing a combination of antibiotics to colistin does not improve mortality in CRAB infections [38,39,40,41], we have summarized in this article the current evidence suggesting the combination of other antibiotics to FDC also does not improve clinical outcomes. Data are still too limited to provide any suggestion as to whether SUL/DUR should be used in combination with other antimicrobials active against CRAB or as active monotherapy.

The literature review from in vitro and in vivo clinical studies suggests that FDC has potent antimicrobial activity against CRAB. Resistance to FDC in *A. baumannii* has been associated with the reduced expression and/or mutations in siderophore receptor *pirA* and *piuA*, and the production of PER- and NDM- β-lactamases. The literature review of clinical studies evaluating the efficacy of FDC therapy against CRAB indicated that FDC is non-inferior to colistin/other treatments for CRAB infections, with a possibly better safety profile. Combination treatment was not associated with improved outcomes compared to monotherapy. Higher mortality rates were often associated with prior patient comorbidities and the severity of the underlying infection.

SUL-DUR, a novel combination of a beta-lactamase inhibitor and non-beta-lactam beta-lactamase inhibitor, restores the susceptibility to SUL and shows excellent antimicrobial activity against CRAB. Although there are limited data regarding clinical studies in the real-life setting on the efficacy of SUL-DUR, current data from the pivotal clinical trial and case reports suggest this antibiotic could be a valuable option in critically ill patients affected by CRAB infections, especially where no other options appear to be effective.

Several open questions remain for the future. Among them are which of the novel antibiotics we presented should be used for diverse types of patients. The evidence is still very limited. With both FDC and SUL-DUR being very well-tolerated agents [67,80], the severity of illness would not be a major determinant. In contrast, the infection syndrome could play a role in clinical practice decision-making. Considering the reduced efficacy of FDC in respiratory infections, SUL/DUR could be a better option in this setting. Indeed, the large majority of patients enrolled in the ATTACK trial [80] had CRAB-related pneumonia. On the other hand, very little evidence has been generated so far on SUL/DUR in the setting of bloodstream infections, where indeed FDC could be preferred. Finally, in light of the broad antimicrobial spectrum, FDC would also be the agent of choice when polymicrobial Gram-negative infections exist or when suspected or documented infection with other XDR microorganisms complicates the clinical course.

Surely, larger clinical studies and/or clinical trials showing adequate power to answer such important clinical questions are strongly awaited.

## Figures and Tables

**Figure 1 antibiotics-12-01729-f001:**
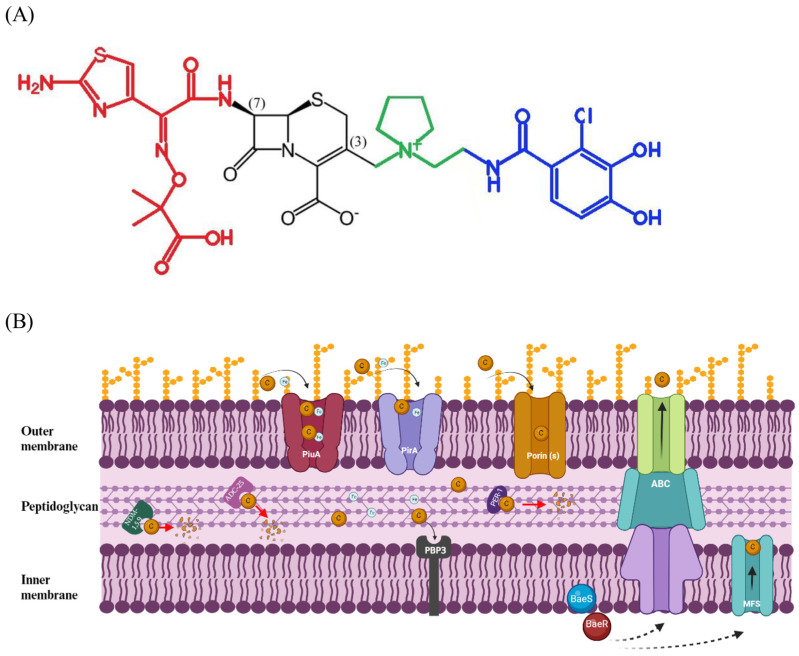
Cefiderocol: structure (**A**), mechanism of action and resistance (**B**). (**A**) Chemical structure of FDC: the C-7 side chain, N-methyl-pyrrolidine group, and the catechol moiety in C-3 chain are highlighted in red, green, and blue, respectively. (**B**) The active transport through PiuA and PirA iron transport systems, the passive diffusion via porin channels, the efflux through ABC and MSF efflux systems, and their modulation by BaeSR two component regulator are shown. The degradation of FDC by NDM 1-5-9, ADC-25, and PER-1 beta-lactamases is displayed. The binding of FDC to penicillin-binding protein 3 (PBP3) target molecule is also shown. Figure created with Biorender.

**Figure 2 antibiotics-12-01729-f002:**
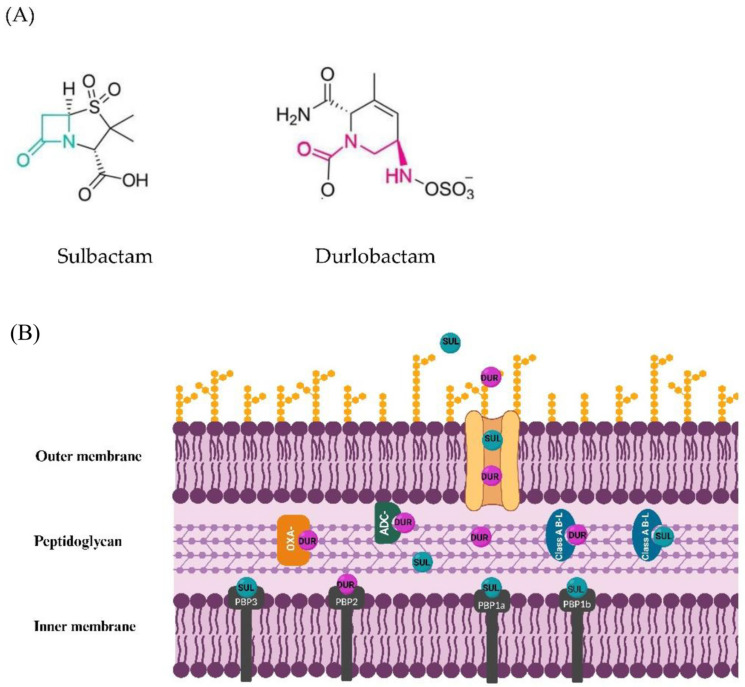
Sulbactam and durlobactam: Structures (**A**), mechanism of action and resistance (**B**). (**A**) Chemical structures of SUL (left) and DUR (right). The beta-lactam ring of SUL and the carbamoylated active site of DUR are highlighted in green and magenta, respectively. The active form of DUR with the cyclic urea opened is shown. (**B**) The passive diffusion of SUL and DUR via porin channels is shown. The binding of SUL to class A β-lactamase (A B-L) and DUR to class A B-L, AmpC-type of class C B-L, and OXA-type of class D B-L is shown. The binding of SUL to penicillin-binding protein (PBP) 1a, 1b, and 3 and DUR to PBP 2 target molecules is also shown. Figure created with Biorender.

**Table 1 antibiotics-12-01729-t001:** Clinical and microbiological outcomes at the test of cure based on types of infections in the CREDIBLE study [36].

Types of Infections	Outcomes							
	Clinical Cure at TOC		Clinical Failure at TOC		Microbiological Eradication at TOC		Microbiological Persistence at TOC	
	FDC	BAT	FDC	BAT	FDC	BAT	FDC	BAT
Nosocomial pneumonia	20 (50)	10 (53)	16 (40)	6 (32)	9 (23)	4 (21)	8 (20)	7 (37)
Bloodstream infections	10 (43)	6 (43)	9 (39)	7 (50)	7 (30)	4 (29)	3 (13)	2 (14)
Complicated urinary tract infections	12 (71)	3 (60)	2 (12)	1 (20)	9 (53)	1 (20)	5 (29)	1 (20)
Overall	42 (53)	19 (50)	27 (34)	14 (37)	25 (31)	9 (24)	16 (20)	10 (26)

Data are N (%). Abbreviations: BAT, best available treatment; TOC, test of cure.

**Table 2 antibiotics-12-01729-t002:** Clinical studies comparing the efficacy of FDC to other treatments in CRAB infections.

Author	Study Design	Site of Infection and Pathogens	No. of Patients	Treatment	Clinical Outcome and Adverse Event (AKI)	Microbiological Outcome	Mortality
Falcone et al. [43]	Single-center observational retrospective study	CRAB: BSI 63.7% VAP 28.5% Other sites 8.1%	124	FDC 47 (37.9%): 15 (31.9%) monotherapy 32 (69.1%) combination	Clinical cure-NA	Microbiological failure FDC 17.4% Colistin 6.8%	All infections 30-day mortality FDC 34% Colistin 55.8% BSI 14-day mortality: FDC 7.4% Colistin 42.3% 30-day mortality: FDC 25.9% Colistin 57.7% VAP no difference [Septic shock, SOFA score associated with mortality]
				colistin-based regimens 77 (62.1%): 12 (15.6%) monotherapy 55 (84.4%) combination	Adverse event FDC 2.1% Colistin 21.1%		
Russo et al. [44]	Single-center, retrospective, observational study	VAP and bacteremia due to CRAB in COVID-19 in ICU	73	FDC 19 (26%) 100% in combination	NA	Microbiological clearance higher in survivors	FDC regimen and FDC+ fosfomycin associated with survival compared to colistin
				colistin-based regimen 54 (74%) 12 (22.2%) monotherapy 32 (77.8%) combination			
Dalfino et al. [45]	Single-center prospective observational study	VAP due to CRAB in COVID-19 and non-COVID-19	90	FDC 40 patients 19 (47.5%) monotherapy 21 (52.5%) + fosfomycin	Clinical failure FDC 25% Colistin 48%	Microbiological failure FDC 30% Colistin 60%	14-day mortality FDC 10% Colistin 38%
				colistin 50 patients all combination Inhaled colistin in both groups	AKI FDC 45% Colistin 47%		
Mazzitelli et al. [46]	Single-center, retrospective, observational study	CRAB: BSI 53 (47.7%) VAP 58 (52.3%)	111	FDC 60 patients 50% monotherapy	Clinical cure FDC 73% Colistin 67% FDC Monotherapy 76.7% Combination 70%	Microbiological eradication FDC 43% Colistin 41% FDC Monotherapy 50% Combination 36.7%	Death FDC 51% Colistin 37% FDC Monotherapy 33.3% Combination 53.3%
				colistin 51 patients all combination	AKI FDC 10% Colistin 25.5%		
Pascale et al. [48]	Multi-center retrospective observational	CRAB BSI 58% LRTI 41%	107	FDC 42 patients monotherapy all cases	Clinical cure 14 days FDC 40% Colistin 36%	Microbiological cure 14 days FDC 28% Colistin 21%	28-day mortality FDC 55% Colistin 58% (High SOFA score risk factor for mortality)
				colistin 65 patients 82% combination	AKI FDC 9.5% Colistin 9.2%		
Rando et al. [47]	Single-center, prospective, observational	CRAB VAP	121	55 FDC monotherapy 21.8% 66 other	Clinical cure-NA	Microbiological failure FDC 53% Other 31%	28-day mortality FDC 44% Other 67% (mortality higher in septic shock and higher SOFA score)
					AKI FDC 9.1% Other 17%		
Bavaro et al. [49]	Single-center, retrospective, observational	CRAB-BSI	118	−43 FDC combination 63%	Clinical cure FDC 60.5% Colistin 41.3%	NA	30-day all-cause mortality FDC 40% Colistin 59%
				−75 colistin combination 96%	Adverse event FDC 2% Colistin 16%		

Abbreviations: AKI, acute kidney injury; BSI, bloodstream infections; CRAB, carbapenem-resistant *Acinetobacter baumannii*; CR GN, carbapenem-resistant Gram-negative; EOT, end of treatment; Cefiderocol, FDC; LRTI, lower respiratory tract infections; IT, urinary tract infection; VAP, ventilator-associated pneumonia.

**Table 3 antibiotics-12-01729-t003:** Clinical studies and case series evaluating efficacy and safety of FDC in CRAB infections.

Study	Study Design	Type Of Infection and Pathogens	Patients	Clinical Outcome and Adverse Events	Microbiological Outcome	Mortality
Clinical studies						
Palermo et al. [57]	Single-center, retrospective observational	CR GN For CRAB: BSI 48.4% HAP 41.9% SSTI 25.8% cIAIs 19.3% cUTI 9.7% Other 12.8%	41 total patients: 31 CRAB	CRAB: Clinical cure at EOT 64.5% Adverse events 4.9%	CRAB: Microbiological eradication at EOT 80.6%	CRAB: 30-day mortality 35.5% VAP 61.5% BSI 46.7%
Calò et al. [50]	Multi-center, retrospective/prospective, observational study	CRAB All types of infections (mostly): BSI 45% Respiratory 40%	38 patients	Clinical failure at EOT 32.5% Monotherapy 27.6% Combination 45.5% (non-significant) Adverse event none	Microbiological failure at EOT 10% Monotherapy 13.8% Combination 0 (non-significant)	30-day mortality 47.5% Monotherapy: 48.3% Combination 45.5%
Giannella et al. [51]	Multi-center, retrospective observational study	*Acinetobacter* spp: Respiratory 65.3% BSI 26.5% Other 8.2%	147 patients 146 *A. baumannii* 1 other *Acinetobacter*	Resolution of infection 39.5% Improved symptoms 12.2% Failure 38.1% Clinical success: BSI 52.2%-75% Respiratory 45.8% Monotherapy 61.2% Combination 49% Adverse event 4.8%	NA	28-day mortality Overall 51% Monotherapy 53.1% Combination 40.8% Survival: septic shock 24.3% Without Septic shock 60.7%
Piccica et al. [52]	Multi-center, retrospective observational	CR GN LRTI 57% IAI 9.2% UTI 8.5% BSI 13.4% ABSSSI 6.3% OTHERS 2.1%	142 patients: 70 monotherapy 72 combination (89 *A. baumannii*)	Clinical cure-NA AKI 38%	Microbiological eradication All pathogens 48.9% Monotherapy 45.8% Combination 52.3%	Survived All pathogens 36.6% Monotherapy 32.9% Combination 40.3% Mortality higher in septic shock (Survival in *A. baumannii* 62.9%)
CASE SERIES						
Bavaro et al. [61]	Case series	XDR GN For CRAB: BSI 70% BSI+ VAP 10% VAP 10% Perihepatic abscess 10%	13 patients: CRAB in 10	CRAB: Clinical cure 70% No adverse event	CRAB: Microbiological eradication 100%	CRAB: Mortality 30%
Corcione et al. [62]	Case series	CR GN VAP + BSI 61.2% BSI 16.7% VAP 11.1% Other 11.1%	18 patients: CRAB in 16: 4 monotherapy 11 combination	All pathogens: Clinical cure 66.7% Monotherapy 75% Combination 64.29% No serious adverse event	All pathogens: Microbiological failure 22.2% Monotherapy 25% Combination 21.43%	All pathogens: 30-day mortality 27.8% Monotherapy 25% Combination 28.57%
Gavaghan et al. [63]	Case series	CR GN For CRAB: Pneumonia 10 UTI 1 Pneumonia + BSI 2 Wound + BSI 1	24 patients: *A. baumannii* in 14	CRAB: Clinical success 35.7% No adverse event	NA	CRAB: Mortality 42.8% 62% of isolates susceptible to FDC
Falcone et al. [60]	Case series	CR GN For CRAB: 6 patients BSI 2 patients VAP	10 patients *A. baumannii* in 8: 7 Monotherapy 1 Combination	CRAB: Clinical success 62.5% No severe adverse event	CRAB: Microbiological failure 25%	CRAB: Mortality 12.5%
Wicky et al. [65]	Case series	DTR GN For CRAB: Mostly VAP	16 patients CRAB in 9: 1 Monotherapy 8 Combination	CRAB: Clinical cure 77.8% Adverse events 66.6% 66.7% (mostly) encephalopathy		CRAB: Death 22.2%
Smoke et al. [64]	Case series	CRAB burn patients 5 BSI 4 BSI/VAP 1 VAP 1 VAT	11 patients: 3 Combination	Clinical cure 36% (FDC resistance 60% of tested isolates)	Microbiological failure 90 days 88% of 8 patients who completed initial treatment	Mortality 27.3%

Abbreviations: ABSSSI, acute bacterial skin and skin structure infection; BSI, bloodstream infections; cIAIs, complicated intra-abdominal infections; CR GN, carbapenem-resistant Gram-negative; CRAB, carbapenem-resistant *Acinetobacter baumannii*; CR GN, carbapenem-resistant Gram-negative; cUTI, complicated urinary tract infections; DTR GN, difficult to treat Gram-negative; EOT, end of treatment; FDC, FDC; LRTI, low respiratory tract infections; UTI, urinary tract infection; SSTI, skin and soft tissue infections; VAP, ventilator-associated pneumonia; VAT, ventilator-associated tracheobronchitis; XDR GN, extensively resistant Gram-negative.

**Table 4 antibiotics-12-01729-t004:** Case description of real-life SUL-DUR use in CRAB infection.

Case Report	Patient	Comorbidities and History Prior to *A. baumannii* Isolation	*A. baumannii* Isolations and Clinical Characteristics	Prior Treatment Failure for *A. baumannii* Infection	Treatment Success	MIC	Outcome
Zaidan et al. [81]	55-year-old female	Comorbidities: Diabetes mellitus hypertension gastric bypass for obesity History prior to *A. baumannii* isolation: COVID-19 pneumonia with RF, mechanical ventilation → clinical improvement, extubation	During hospitalization, after improvement of COVID-19 pneumonia: → refractory hypoxia, intubation, vasopressor support → respiratory culture PDR *A. baumannii* Wild-type *P. aeruginosa*	Empiric meropenem+vancomycin After AST results meropenem + ampicillin/SUL	SUL-DUR 2 gr q6h + FDC 2 gr q8h	SUL-DUR 4 mg/L FDC 0.5 mg/L	Improvement after 72 h 14-day course treatment → tracheostomy- → discharged long-term care facility → discharged home
Holger et al. [82]	50-year-old male	Pulmonary embolism + infarction → intubation, thoracotomy, partial decortication	Bronchoalveolar lavage *A. baumannii* meropenem S → thoracotomy, RLL resection (abscess), decortication → Pleural tissue *A. baumannii* I to colistin → 45 days later, persistence of *A. baumannii* which developed resistance to FDC, eravacycline	Empiric piperacillin/tazobactam + vancomycin After AST results meropenem tigecycline colistin + meropenem FDC eravacycline FDC + tigecycline	SUL-DUR 2 gr q8 + meropenem 1 gr q6	SUL-DUR MIC 8 mg/L alone 4 mg/L with meropenem	13-day treatment Clinical cure Discharged home
Tiseo et al. [83]	Young female	Burn injury 45% of total body surface area → intubated	Central line BSI CRAB R to FDC, S to colistin → skin lesion CRAB + CR *P. aeruginosa* → BAL: CRAB R to FDC and colistin + CR *P. aeruginosa*	Colistin + tigecycline	SUL-DUR + colistin (for *P. aeruginosa*)	SUL-DUR MIC 1.5 mg/L	12-day regimen Clinical improvement Microbiological eradication of CRAB Alive at 30 days

Abbreviations: AST, antibiotic susceptibility testing; RF, respiratory failure; BSI, bloodstream infection; BAL, bronchoalveolar lavage; CR, carbapenem-resistant; CRAB, carbapenem-resistant *A. baumannii*; S, susceptible; I, intermediate; RLL, right lower lobe; R, resistant; SUL-DUR, SUL/DUR; FDC, FDC.

## Data Availability

No new data were created or analyzed in this study. Data sharing is not applicable to this article.

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
