# Peer review of "Cefiderocol and Sulbactam-Durlobactam against Carbapenem-Resistant Acinetobacter baumannii"

_antibiotics, 2023, doi:10.3390/antibiotics12121729_

Round 1

Reviewer 1 Report

Comments and Suggestions for Authors

The review is devoted to a description of the in vitro activity and clinical effectiveness of two new antibiotics intended for the treatment of infections caused by carbapenem resistant Acinetobacter spp. –   Cefiderocol and Sulbactam-Durlobactam. The authors conducted a detailed analysis of the literature on the topic. However, some issues require additional coverage and clarification.

·        Readers not sufficiently familiar with the mechanisms of iron transport into the bacterial cell would be interested in receiving more detailed information on this issue (lines 69-73).

·        When describing the mechanisms of resistance to cefiderocol, it is advisable to consider the problems of heteroresistance, crossresistance to ceftazidime/avibactam and other beta-lactams, and the role of PBP modifications.

·        Although the problem of emergence of resistance during therapy is briefly mentioned it is advisable to consider it in more detail.

·        It should be noted that diazabicyclooctanones inhibitors exhibit variable activity against beta-lactamases of the class D beta-lactamases (line 347).

·      Also, it should be mentioned that sulbactam only slightly inhibit CTX-M-15, SHV-5, and KPC-2 class A beta lactamases (line 351).

Author Response

Response to Reviewer 1

Comments and Suggestions for Authors

The review is devoted to a description of the in vitro activity and clinical effectiveness of two new antibiotics intended for the treatment of infections caused by carbapenem resistant Acinetobacter spp. –   Cefiderocol and Sulbactam-Durlobactam. The authors conducted a detailed analysis of the literature on the topic. However, some issues require additional coverage and clarification.

Response: We thank reviewer 1 for skillfull editing of the manuscript.

  • Readers not sufficiently familiar with the mechanisms of iron transport into the bacterial cell would be interested in receiving more detailed information on this issue (line 69-73).

Response: The mechanisms of iron transport into the bacterial cells has been detailed (page 2, line 81 of revised manuscrtipt).

  • When describing the mechanisms of resistance to cefiderocol, it is advisable to consider the problems of heteroresistance, crossresistance to ceftazidime/avibactam and other beta-lactams, and the role of PBP modifications.

Response: The problems of heteroresistance, crossresistance to ceftazidime/avibactam and other beta-lactams, and the role of PBP modifications have been discussed (page 3, lines 139-142; page 4, lines 166-167; page 4, lines 168-176).

  • Although the problem of emergence of resistance during therapyis briefly mentioned it is advisable to consider it in more detail.

Response: In vivo emergence of resistance or reduced suceptibility to FDC during therapy was consider in more detail (page 11, lines 355-376).

  • It should be noted that diazabicyclooctanones inhibitors exhibit variable activity against beta-lactamases of the classD beta-lactamases (line 347).

Response: The information that “DUR exhibits variable activity against class D beta-lactamases, the highest activity being found for OXA-48 with respect to OXA-10, OXA-23 and OXA-24” has been included in the revised manuscript (page 14, lines 401-403).

  •     Also, it should be mentioned that sulbactam only slightly inhibit CTX-M-15, SHV-5, and KPC-2 class A beta lactamases (line 351).

Response: The information that sulbactam “exhibits weak intrinsic inhibitory activity against CTX-M-15, SHV-5, TEM-1 and KPC-2 class A serine β –lactamases, but not class C and D beta-lactamases” has been provided (page 14, lines 405-407 of the revised manuscript).

Reviewer 2 Report

Comments and Suggestions for Authors

I have reviewed the manuscript entitled “Cefiderocol and Sulbactam-Durlobactam against Carbapenem Resistant Acinetobacter baumannii” submitted for possible publication in the journal “Antibiotics”. The authors have done great efforts in compiling all the relevant literature to address to problem. The title is well elaborated and representing the content of manuscript. The manuscript can be proceeded further for possible publication after addressing some comments and suggestions. My specific comments are:

1.     The manuscript should be checked for English proofreading as some of the sentences are meaningless and there is no continuity.

2.     I would like to recommend the authors to add conclusive remarks in the abstract.

3.     Line 17-18: Mention here about the approval from which regulatory body.

4.     Line 28: remove “glucose”.

5.     Line 32-33: Divide the sentence into 2 separate sentences.

6.     Line 50-53: Mention here also about cefiderocol as a part of objective.

7.     I would like to suggest the authors to add about current issues and future direction to address the AMR problem among A. baumannii isolates.

8.     The conclusion section needs to be revise as it is a review but not the original article. The authors should revise their statements e.g., The literature review from invitro and in vivo studies” etc. etc.

Comments on the Quality of English Language

The manuscript should be checked for English proofreading as some sentences are meaningless and have no continuity.

Author Response

Response to Reviewer 2

Comments and Suggestions for Authors

I have reviewed the manuscript entitled “Cefiderocol and Sulbactam-Durlobactam against Carbapenem Resistant Acinetobacter baumannii” submitted for possible publication in the journal “Antibiotics”. The authors have done great efforts in compiling all the relevant literature to address to problem. The title is well elaborated and representing the content of manuscript. The manuscript can be proceeded further for possible publication after addressing some comments and suggestions.

Response: We thank Reviewer 2 for her/his positive evaluation of the manuscript.

My specific comments are:

  1. The manuscript should be checked for English proofreading as some of the sentences are meaningless and there is no continuity.

Response: The manuscript has been accurately checked for English proofreading and some sentences have been corrected to make the English language more fluent.

  1. I would like to recommend the authors to add conclusive remarks in the abstract.

Response: The conclusive remarks were included in the abstract (page 1, lines 22-30).

  1. Line 17-18: Mention here about the approval from which regulatory body.

Response: The information that cefiderocol was “approved by the Food and Drug Administration for the treatment of A. baumannii infections” has been provided (page 1, line (18).

  1. Line 28: remove “glucose”.

Response: The word “glucose” was removed (page 1, line 37).

  1. Line 32-33: Divide the sentence into 2 separate sentences.

Response: The original sentence was divided into 2 separate sentences ans suggested (page 1, lines 40-42).

  1. Line 50-53: Mention here also about cefiderocol as a part of objective.

Response: The cefiderocol was mentioned as a part of objective as suggested (page 2, lines 60-63).

  1. I would like to suggest the authors to add about current issues and future direction to address the AMR problem among A. baumanniiisolates.

Response: Following reviewer’s suggestion, the “Conclusions” section was modified into “Conclusions and future directions” (page 19, lines 550-597 of revised manuscript).

  1. The conclusion section needs to be revise as it is a review but not the original article. The authors should revise their statements e.g., The literature review from invitro and in vivo studies” etc. etc.

Response: The statements have been revised as suggested (page 19, lines 570-578).

Comments on the Quality of English Language

The manuscript should be checked for English proofreading as some sentences are meaningless and have no continuity.

Response: The manuscript has been accurately checked for English proofreading and some sentences have been corrected to make the English language more fluent.

Reviewer 3 Report

Comments and Suggestions for Authors

The article focusing on the antimicrobial resistance mechanism of A.baumannii shares informative facts. The article will be helpful to researchers working/dealing with hospital acquired A.baumannii infection. When addressing to prevent or control the rise of MDR its very important to understand the complex mechanism exhibited by the bug.

Author Response

Response to Reviewer 3

Comments and Suggestions for Authors

The article focusing on the antimicrobial resistance mechanism of A.baumannii shares informative facts. The article will be helpful to researchers working/dealing with hospital acquired A.baumannii infection. When addressing to prevent or control the rise of MDR its very important to understand the complex mechanism exhibited by the bug.

Response: We thank reviewer 3 for her/his positive evaluation of the manuscript.

Reviewer 4 Report

Comments and Suggestions for Authors

Comments on the Quality of English Language

Author Response

Response to Reviewer 4

Antibiotic use is lifesaving despite its antibiotic resistance risk. Antibiotic against CARB in critically ill patients is a must. Unfortunately, it is also used for nonsevere infection disease.

The author should consider a well-judged antibiotic use (only included study CARB in critically ill patients) because these antibiotics are a new antibiotic (WHO Reserve category). Therefore, the author should present more detailed benefits and risk of monotherapy and combination therapy. Usually, combination antibiotic therapy is used for a broad antibiotic spectrum.

Response: Recommendation for FDC and SUL/DUR use and benefits and risk of monotherapy and combination therapy has been included throughout the text (page 11, lines 374-376; page 14, lines 4010-403; page 19, lines 550-597).

Table 2, 3, 4 are difficult to understand. Fig 1 and Fig 2 describe the site of action, can author add information about resistance mechanism, especially that related (interaction with) β-lactamases (page3, subsection 2.3 Mechanisms of resistance; page 13, subsection 3.3) and/or gene resistance (genotyping). Which factor is dominant for antibiotic resistance (Cefiderocol, sulbactam - durlobactam)?

Response: Tables 2, 3 and 4 have been modified following reviewer’s request. To avoid confusion, additional information on the mechanisms of resistance were not added to Figures 1 and 2. Instead, additional information on the resistance mechanisms to FCD and SUL/DUR were detailed in the main text of the manuscript (pages 4 and page 14 of revised manuscript).

In conclusion, what antibiotic (FDC or SUL-DUR) for what type of patients (condition, type and severity of infection diagnosis) that the author recommend in clinical practice ?

Response: Recommendation for FDC and SUL/DUR use in the clinical pratice have been discussed in “Conclusions and future directions” section (page 19, lines 550-597).